# Hygiene Knowledge and Practices of Portuguese Hunters Using Wild Boar Meat for Private Consumption

**Ana Carolina Abrantes** [1,*] , **João Canotilho** [2] **and Madalena Vieira-Pinto** [1,3,4,*] 

1. Animal and Veterinary Research Centre (CECAV), Trás-os-Montes e Alto Douro University (UTAD), 5000-801 Vila Real, Portugal
2. Reprovet—Serviços Veterinários, 6300-749 Guarda, Portugal
3. Department of Veterinary Sciences, Trás-os-Montes e Alto Douro University (UTAD), 5000-801 Vila Real, Portugal
4. AL4AnimalS—Associate Laboratory for Animal and Veterinary Sciences, 5000-801 Vila Real, Portugal
* Correspondence: carolina.psca@gmail.com (A.C.A.); mmvpinto@utad.pt (M.V.-P.)

**Simple Summary:** Hunters who dress and eviscerate carcasses for self-consumption have no access to hygienic conditions like those present in modern slaughterhouse facilities. Together with the behaviour of self-consumption without previous initial examination or inspection of the hunted wild boar's carcasses after each driven hunt, some poor evisceration, handling, and hygiene practices occur. Bad practices such as too long a time between shooting, evisceration, and cooling; wrong handling methods of the carcasses when faecal contamination is observed; and not using proper equipment are common among Portuguese hunters. These are risky practices that lead to food insecurity and the acquisition of foodborne diseases from this self-consumption of wild boar meat. Added to all this is the fact that hunters do not recognise lesions compatible with zoonotic infections when eviscerating and handling the carcasses. There is a lack of information and training on this issue. It is necessary to join efforts (hunters' associations, scientific community, and public health authorities) and develop strategies to increase the knowledge of best practices among this population at risk.

**Abstract:** The microbiological contamination of wild boar meat depends on the hygiene practices that hunters apply during its preparation, from the point of collection to its refrigeration. This study assesses Portuguese hunters' knowledge of hygiene practices when handling wild boar carcasses that can jeopardise meat safety. A general structured survey entitled "Private consumption of game meat and good hygiene practices" was distributed to Portuguese hunters. Of the 206 respondents, 95% use wild boar meat for private consumption without prior inspection or initial examination by a veterinarian. This study also revealed that the vast majority of respondents have several risky practices that can compromise the safety of wild boar meat consumed (evisceration, handling, refrigeration, and transport). It is also evident that there is a lack of knowledge related to recognising lesions compatible with zoonotic infections in the hunted animals. These inappropriate knowledge and practices can pose a risk to hunters (occupational zoonotic health) and consumers (foodborne diseases). To reduce this risk, hunters need to be trained and informed about proper game meat handling practices.

**Keywords:** food safety; game meat; good hygiene practices; public health

## 1. Introduction

An important issue for the food market today is the concept of organic food, along with local or "zero-mileage" food: The consumer is more willing to promote sustainable food products and local economies. Game meat fits into this sustainable and organic food when obtained through local hunting activity [1–3].

Some European Union (EU) regulations (EU Reg 178/2002, 852/2004, 853/2004, and 627/2019) legislate responsibilities, traceability, and game meat safety, ensuring the same level of control granted to domestic animal meat [4].

Like meat from slaughtered domestic animals, meat from wild game can lead to foodborne infections. Therefore, the presence of foodborne pathogens is important in the evaluation of the hygienic status of this type of meat [5]. However, no specific microbiological criteria exist for game meat [5]. Still, it is known that contamination is dependent on several factors (intrinsic, extrinsic, and bad practices of handling) [6,7], and it is not surprising that the microbiological quality of game meat has also been found to be highly variable [5,7–11].

Some specific steps are particularly stressed as they are crucial because these wild animals are shot, bled, and eviscerated in wild/external environments. Meat hygiene and microbiological contamination depend on how animals are killed (e.g., different hunting methods and shot location in the animal's body), dressed, and handled from the collection to the chilling point. Previous studies explored the microbiological quality of game meat [5,7,12,13]. However, there is scarce information regarding the microbial communities present in wild boar meat.

If animals are correctly shot and adequately dressed, the microbial contamination of fresh carcasses may be minimal [2,5,7]. Besides this, the hunting process and the subsequent handling (hygiene and temperature, for example), transportation, and slaughtering processes are very critical steps, especially in warm seasons [14,15]. For instance, the time between shooting and evisceration must be as short as possible. After a trained person has evaluated the absence of macroscopic lesions during the initial examination, the carcass must be cooled to a maximum of 7 °C in a short time. Skinned carcasses must be stored separately from not-skinned ones, and skinning must be carried out correctly by changing the knives or washing them frequently; it is also fundamental to avoid the contamination of muscles with gastrointestinal content possibly left in the abdomen and, at the same time, prevent contamination from the skin and fur [5,6,16,17]. Food hygiene regulations do not recommend washing the carcass with water as it could spread bacteria [13].

Several studies suggest that some of the practices commonly used by hunters for the evisceration, handling, and storage of game meat are not carried out correctly [18]. Additionally, a considerable amount of large game meat is used for private consumption. Nevertheless, an official post-mortem veterinary inspection of the carcass in a slaughterhouse or a game-handling establishment is required to declare that meat is suitable for human consumption if placed on the market.

In areas where hunting is a traditional activity, such as the Iberian Peninsula, where Portugal is located, there are several studies on the prevalence of pathogens present in wild boar that can cause potential foodborne infections. It is known that infections such as *Toxoplasma gondii*, *Erysipelothrix rhusiopathiae*, hepatitis E, *Trichinella* spp., *Campylobacter* spp., *Salmonella* spp., and other foodborne pathogens are present in Iberian wild boar populations but with different prevalence values and endemicity levels. For example, colibacillosis is one of the foodborne infections with the highest prevalence (95% in the Spanish wild boar population according to a study conducted in 2017) [19].

It is crucial to ensure game meat's hygiene and safety standards to mitigate the potential risk to the consumer. In view of this, all regulations consider training food business operators in all the rules deemed essential [2]. It is also evident from various studies that hunters, "real operators" present on the spot after each driven hunt, lack knowledge about these issues. Poor hygiene practices; the lack of knowledge about some of the diseases that can affect large game meat, especially zoonotic diseases; and the lack of disease prevention strategies are highlighted as some of the greatest risks of self-consumption of game meat [18,20].

The present study assesses Portuguese hunters' knowledge of good hygiene practices in handling wild boar carcasses that may compromise the safety of the meat.

## 2. Materials and Methods

A general structured survey entitled "Private consumption of game meat and good hygiene practices" was distributed among Portuguese hunters. The sample (*n* = 206) was obtained with the objective of reaching as many hunters as possible during the 2020/21 and 2021/22 hunting seasons.

Two survey versions were distributed: an online version and a paper version distributed during hunting events. The survey was randomly distributed during the driven hunts, and all hunters filled it out willingly. The surveys were anonymous, and before they were filled in, each hunter gave informed consent, in accordance with the Data Protection Act. The survey was validated beforehand by distributing them among a group of ten hunters.

The following variables were collected for analysis in this study: being a hunter, whether the initial examination course for large game products was taken, contact with hunted animals' carcasses, food safety habits, hygiene practices in evisceration and initial examination, recognition of lesions compatible with zoonotic diseases, and perception of the zoonotic risk and receptiveness to obtain more information regarding the survey topic. Personal and demographic data were not collected, and informed consent was obtained from all hunters.

Specific questions (Table 1) were raised in the survey regarding (1) hygiene practices, asking the time between shot, evisceration, and chilling; the use of gloves in the evisceration; and washing or cutting carcass parts to remove faecal contamination; (2) the recognition of lesions compatible with zoonotic pathogens, with the presentation of images of lesions compatible with hydatidosis, *Erysipelothrix rhusiopathiae* infection, and Tuberculosis; and (3) the perception of the importance of evisceration and initial examination at the collection point and the necessity of training on good practices.

**Table 1.** Summary of the questions applied in the questionnaire, answer options, and percentage of answers from the 206 hunters surveyed.

| Questions | Responses | Data (%) |
|---|---|---|
| **(1)  General questions about private consumption:** | | |
| "Do you self-consume wild boar meat?" | Yes | 95% |
| | No | 5% |
| "Will you share this wild boar meat with your family and friends? | Yes | 99% |
| | No | 1% |
| "Do you eviscerate the wild boar you hunt at the collection point?" | Yes | 65% |
| | No | 35% |
| **(2)  Questions about hygiene practices:** | | |
| "On average, how long after hunting did you eviscerate the carcasses?" | Up to 3 h | 48.1% |
| | Between 3 and 6 h | 41.7% |
| | More than 6 h | 10.2% |
| "On average, after eviscerating, how long does it take before you put the carcasses in to chill?" | Up to 3 hours | 52.8% |
| | Between 3 and 6 h | 26.4% |
| | More than 6 h | 20.8% |
| "To proceed with the evisceration of carcasses, do you always wear gloves?" | Always | 47.1% |
| | Never | 26.2% |
| | Sometimes | 26.7% |

**Table 1.** *Cont.*

| Questions | Responses | Data (%) |
|---|---|---|
| **(2)    Questions about hygiene practices:** | | |
| "When preparing the carcass and seeing faecal contamination, do you usually wash the carcass to remove it?" | Always | 67.9% |
| | Never | 14.2% |
| | Sometimes | 17.9% |
| "If not, instead of washing, do you usually remove the faecal contaminated parts of the carcass?". | Always | 47.2% |
| | Never | 8.5% |
| | Sometimes | 29.2% |
| | No response | 15.1% |
| **(3)    Recognition of lesions compatible with zoonotic pathogens:** | | |
| "Have you ever seen this lesion on the liver of the animals you hunt?" (question with image compatible with hydatidosis) | Yes | 30.1% |
| | No | 69.9% |
| "Have you ever seen these lesions on the skin (purple, lozenge-shaped patches) and heart (warts on the inside of the heart) of the wild boar you hunt?" (question with images compatible with *Erysipelothrix rhusiopathiae* infection) | Yes | 18% |
| | No | 82 % |
| "During evisceration, have you ever seen this type of lesion on the carcass (purulent material in the mandibular and mesenteric lymph nodes)?" (question with images compatible with Tuberculosis) | Yes | 22.3% |
| | No | 77.7% |
| "When you see these lesions on the carcasses, how do you usually proceed?" | Only discard the affected parts. | 82% |
| | Discard the complete carcass. | 10% |
| | Usually, do nothing. | 0% |
| | Look for the opinion of the Vet. | 39% |
| "How do you dispose of by-products/waste (skin, guts, bones, etc.)?" | Burial | 69% |
| | Abandoned in the field | 35% |
| | Feeds other animals | 2% |
| | Put it in the trash | 0% |
| | Burn | 3% |
| | Collected by UTS * | 0% |
| **(4)    Perception of good practices necessity and training:** | | |
| "Have you taken the 'Large game initial examination course' for hunters?" | Yes | 44.3% |
| | No | 55.7% |
| "Do you think it is important to eviscerate a carcass and perform the initial examination at the collection point after each driven hunt?" | Yes | 72% |
| | No | 4% |
| | There are a few problems. | 24% |
| "Do you think it is important for the veterinarian to be present at the collection point to perform the initial examination?" | Yes | 99% |
| | No | 1% |
| "Would you like to have more training in good hygiene practices and initial examination of the large game?" | Yes | 95% |
| | No | 5% |

\* UTS.

## 3. Results and Discussion

Two hundred and six hunters (*n* = 206) responded to the survey from across the nation, comprising 140 participants from northern Portugal (68%), 39 from central Portugal (19%), and 27 from the southern region (13%). The majority of the respondents (95%) eviscerate wild boar carcasses for private consumption at collection points or in their own homes. From these, more than 50% do not eviscerate the hunted animals at collection points. All of the 206 participants in the survey pointed out that an initial examination by a veterinarian is also not a common practice in the driven hunts in which they participate. The main results are presented in Table 1, underlining the fact that the current practices the hunters acknowledged can pose a risk to their health.

Hunting is an activity firmly rooted in Portuguese culture and important to the economy. Wild boar is the most hunted large game species, and the private consumption of its meat is a common practice. Consumption behaviour similar to that found among Portuguese hunters has also been described by Spanish hunters, as indicated in the work by Lizana et al., (2022) [18]. In our study, 95% of the respondents reported that the game they hunted was used for private consumption, and they also stated that this meat was consumed by family and friends. Together with this behaviour of private consumption without previous initial examination on the spot, some risky evisceration, handling, and hygiene practices occur [21,22]. The lack of initial examination of carcasses poses a health risk not only to hunters but to a broad population (family, friends, etc.), who probably have scarce information about the risks involved in large game meat consumption [21–23]. Cases of foodborne transmission of some zoonotic agents have been reported concerning the self-consumption of wild boar meat without sanitary inspection and the production of derived meat products [22,23].

In the case of Portugal, a major problem is that many hunters travel throughout Portugal to hunt. Thus, the place of origin of the hunted animal (with its specific epidemiological health characteristics) might be several kilometres away from the location where the animal will be eviscerated and cut. This increases the possibility of the dispersion of diseases to other regions of the country with different epidemiological characteristics. To hunters, there is a potential risk of foodborne exposure during all stages of the hunting process, namely harvest, carcass dressing, storage, and consumption [21,24], and this risk could be higher in cases of private consumption [25]. It is essential to avoid the "human factor" in the dispersion of diseases and the microbiological contamination of wild boar carcasses handled by local consumers, with basic meat hygiene procedures and handling as well as personal hygiene [25,26].

In terms of carcasses' handling and hygiene practices, a large number (48.1%) of hunters affirmed that they eviscerate animals up to 3 hours after shooting (which is the maximum hours recommended in the literature [27]), and 52.8% chill carcasses up to 3 hours after evisceration. To eviscerate carcasses, 26.2% of the hunters never use gloves, whereas 47.1% always use gloves, and 26.7% sometimes use gloves. Using gloves has been described as one of the most effective ways of preventing hunters from acquiring infectious diseases when handling and eviscerating the carcasses of the animals they hunt. The acquisition of zoonotic diseases such as Brucellosis, Hepatitis E, *Erysipelothrix rhusiopathiae* infection and Tuberculosis can be prevented by wearing gloves [6,7,14,18,19,28].

With this preliminary evaluation, it can be concluded that hunters commonly use bad hygiene practices that reduce the quality of the meat and pose a risk to consumers [28]. Carcasses that are not eviscerated at collection points after each driven hunt are then transported by hunters to their homes (some outside the hunt region), where they are then eviscerated. This significantly increases the time between hunting and evisceration. The evisceration of animals should be carried out as soon as possible after slaughter, as this allows for faster cooling of the carcass, as well as the observation of lesions, and it reduces the speed of deterioration [4,29–31]. The scientific literature states that the bacteriological quality of the carcasses during harvest, evisceration, initial examination, and transportation is unknown [26]. The importance of four critical steps has been highlighted for the hygienic

quality of the meat: field dressing/evisceration, cutting and processing, the disposal of edible organs and carcass parts, and refrigeration [5,21,32]. According to Cenci-Goga et al., (2021) [32], refrigeration is a crucial issue. Still, it is often carried out late and with logistical difficulties due to the distance from the hunting location to the refrigeration location, delayed handling procedures, and conditions intrinsic to the animal, such as its weight. Timely refrigeration is a crucial parameter to stop the microbial contamination of carcasses, and unskinned carcasses should never be refrigerated [16,33].

In this study, more than 50% of the respondents wash carcasses to remove faecal contamination (67.9%); however, if not washed, 47.2% of the hunters always remove the affected parts of the carcass that are dirty with faecal contamination. Regarding washing carcasses, European regulation states that this procedure should not be implemented, as the correct procedure is the removal of the affected parts of the carcass [16,20]. This poor practice may pose a risk to the consumer if zoonotic bacteria are present, since during washing, bacteria spread throughout the carcass with water and respective aerosols [20].

When addressing the issue of lesion recognition, in the presented questions, we aimed to determine whether the hunters had ever observed lesions like the images provided in the questionnaire, which were representatives of some zoonoses found in wild boar throughout Europe [19]. Some of the surveyed participants (30.1%) affirmed that they recognise and have seen hydatidosis in a liver previously, 18% have seen *Erysipelothrix rhusiopathiae* infection, and 22.3% recognise tuberculosis-like lesions. A large proportion (82%) of the respondents discard only the part of the carcasses where the lesion is found, and 10% discard the complete carcass. However, 39% of the hunters admitted to deferring to a veterinarian's opinion.

As there is a specific course in Portugal based on European regulations for properly conducting an initial examination of large game [2,25,34], hunters can attend this course and become a "trained person" with sufficient competencies to perform initial examinations of large game and acquire knowledge about diseases of large game, and they can then correctly identify lesions during the hunt scenario [2,25]. Only a small number of surveyed hunters indicated that they had already seen these lesions in loco during evisceration and initial examination, and not all respondents knew what procedures to adopt in case of compatible lesions. At present, tuberculosis in large game is considered one of the most frequently reported diseases, but it is not the only one identified in Portugal. Other diseases in Portuguese large game that pose health risks to humans or animals include hepatitis E, hydatidosis, *Erysipelothrix rhusiopathiae* infection, Brucellosis, Toxoplasmosis, and Aujeszky's disease [19,35–37]. To manage health risks to humans and animals, the recognition of lesions compatible with different diseases is crucial during the initial examination, in addition to knowing the correct procedures to follow. These results lead us to the conclusion that it is necessary to train hunters to be able to identify lesions and proceed in such a way that guarantees their safety as well as the safety of all those who come into contact with wild boar carcasses since the percentage of hunters who observed the presented lesions in this study is much lower than the findings of various scientific studies published based on data from Portugal [19,36,37].

Regarding the form of elimination of by-products, in this survey, it was observed that the most common form of elimination is the burial of by-products, accounting for 69% of the answers. Notably, 35% of the hunters interviewed opt to abandon them in the field, 3% burn them, and 2% use them as food for their dogs. These answers show that some hunters are still in the habit of abandoning their offal in the field or offering it to dogs as food. The percentage of responses in this survey differs from those presented in the study by Lizana et al., (2022) [18]. In their study, also on the Iberian Peninsula (in Spain), 33.2% of hunters replied that they leave by-products directly in the field, and only 11.1% bury them. In our study, burial was the most common method. In the Spanish study, more than 40 per cent of the hunters also admitted to throwing the remains of hunted animals in the rubbish bin near their homes, while in our study, none indicated this practice [18]. This type of behaviour, accounting for almost a third of the answers, represents a risk to

human and animal health (domestic or wild). This method of carcass disposal allows for the transmission of various diseases. The correct method for the disposal of these by-products is proper elimination in a UTS (a by-product treatment unit) [38].

Based on the responses in the section about risk perception, hunters recognise the importance of evisceration and initial examination at the collection point (72%), and when questioned about the significance of the presence of a veterinarian at collection points to perform an initial examination after each driven hunt, only two hunters answered negatively, which shows that most hunters (99%) recognise the importance of veterinarians' presence. However, 24% of the hunters who responded to the questionnaire stated that in loco evisceration causes problems, pointing to two main reasons for their reluctance in adopting such a procedure: (i) the packaging and transport of wild boar is more complicated, and (ii) the evisceration of wild boar would make it impossible to roast the carcass. These problems of transport logistics that the hunters indicated increase the contamination problems of the carcass since they are not eviscerated in a short time (<2–3 h), and consequently, they are not stored and refrigerated in a short time either [27].

Notably, 95% of the hunters would like to have more training on good hygiene practices and initial examination, even though 44% have already attended the course of initial examination for large game prepared specifically for hunters. Existing research affirms that the microbiological quality of meat is lower when it is handled by an untrained hunter [29]. Thus, in the future, more attention should be paid to the training of hunters to avoid these incorrect hygiene practices that can be a risk to consumer health from the self-consumption of wild boar meat.

There is a long way to go to reduce the risk of zoonotic infection for hunters and processors since the focus should be on education for enhanced knowledge, training, and sensitivity to change procedures and thus practice better handling, gutting, and consumption of game meat. Good handling practices include early evisceration (as soon as possible after the hunt without the need for transporting uneviscerated animals over long distances); the correct refrigeration of carcasses; the use of personal protective equipment, in particular, specific clothing and knives and gloves; and the knowledge of basic procedures for conducting an initial examination of large game pieces to more effectively recognise possible lesions compatible with infections by zoonotic agents [17,18,20,34,35].

## 4. Conclusions

With these results, we conclude that there is still a long way to go in hunters' training so that good hygiene practices and initial examinations are performed correctly and that they recognise the risk of certain zoonoses to public health.

These inappropriate practices (for example, poor evisceration methods and inappropriate handling of carcasses, too long a time between evisceration and cooling, and failure to recognise lesions that pose a zoonotic risk) can pose a risk to hunters (occupational health) and consumers (foodborne diseases). To reduce this risk, promoting the knowledge of good practices for handling game meat is necessary. Hunters themselves are the ones who recognise that there is a lack of information and training in this area and are willing to acquire more knowledge. It is, therefore, necessary to join efforts (hunter associations as well as the scientific community and public health authorities) and develop strategies to enhance the knowledge of this population at risk, especially hunters who use wild boar for self-consumption.

**Author Contributions:** Conceptualisation, A.C.A., J.C. and M.V.-P.; methodology, A.C.A. and J.C.; formal analysis, A.C.A.; investigation, A.C.A. and J.C.; data curation, A.C.A.; writing—original draft preparation, A.C.A.; supervision, M.V.-P. All authors have read and agreed to the published version of the manuscript.

**Funding:** This work was supported by projects UIDP/00772/2020 and LA/P/0059/2020 funded by the Portuguese Foundation for Science and Technology (FCT).

**Institutional Review Board Statement:** Anonymous questionnaires about opinions, satisfaction with several matters, etc., do not need approval. However, a prelude is needed informing about the goal of the investigation, the voluntariness of participation, and the anonymous treatment of data or about the database containing the information. Finally, it must be indicated that respondent volunteers accept their participation in the study (informed consent).

**Informed Consent Statement:** Informed consent was obtained from all subjects involved in the study.

**Data Availability Statement:** The raw data from this study can be obtained by requesting it from the corresponding authors.

**Conflicts of Interest:** The authors declare no conflict of interest.

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
