# Peer review of "Hygiene Knowledge and Practices of Portuguese Hunters Using Wild Boar Meat for Private Consumption"

_zoonoticdis, doi:10.3390/zoonoticdis3040025_

Round 1
Reviewer 1 Report
Comments and Suggestions for Authors
Dear authors,
A good work in general, but some aspects should be reviewed:
- Lines 29 to 32: Could be interesting to consider the citation of
https://doi.org/10.4195/nse.2013.0024
- Lines 78 to 84:
· Is there 206 surveyed hunters enough to consider this study representative to the hunter population in Portugal? How many people practice hunting in Portugal?
· Please provide a complete copy of the questionnaire, no only a summary, as shown in Table 1 (maybe as an Annex). To make possible the replicability of your survey.
· Have you consult other previous surveys to elaborate your questionnaire?
· How did you avoid poll crashing in the online version?
- Lines 100-101: "Two hundred and six hunters (n=206) have responded to the survey, from all over Portugal" Could you provide a map with the origin of the answers to assure this.
- Lines 103-104:"In all of the 206 responses, the initial examination by a veterinarian is also not a common practice". Indicate a percentage.
- Line 120: Could be convenient to add a reference closer to the geographic area of the survey https://doi.org/10.1111/zph.12343
- Lines 171-175: The trained person course it is mentioned. Are there differences in the answers provided by hunters who have completed the course in comparison with those who haven't completed it?
- Line 182: [17,34-35]"." A dot "." is lacking.
- Line 188: "obtained in various scientific studies published with data from Portugal [17,34-35]". Reference 35 is not a scientific study, but a guide. And it comes from Spain, with no mention about Portugal in the text.
-Lines 189-197: It is interesting to add a comparison with the results achieved by https://doi.org/10.3390/foods11030368
- Line 197: There is a reference, identified as 36, that does not appear in the bibliography.
- Line 287: Problem with the format in reference 17.
Conclusions (lies 226 to 237): some conclusions are similar to those exposed by https://doi.org/10.3390/foods11030368 so should be cited.
Author Response
Ana Carolina Abrantes
University of Trás-os-Montes and Alto Douro, (UTAD)
Quinta de Prados
5001-801 Vila Real, PORTUGAL
carolina.psca@gmail.com
Editor of Zoonotic diseases Journal
Vila Real, September 14th 2023
Please find enclosed the resubmission of the manuscript entitled “Knowledge and Hygiene Practices among Portuguese hunters who practice self-consumption of wild boars” renamed as a reviewer’s suggestion.
After an exhaustive reformulation of the text and improvement of the English, we think that the document is now more clarified.
I would add that all the questions raised by the reviewers were answered and the text was adjusted in order to clarify the confusing ideas that were exposed. Attached is a document with individual responses to each point pointed out by the reviewers.
Thank you in advance for the effort of everyone and especially the reviewers for the comments, we hope that the joint effort has substantially improved the document in order to gather conditions to be accepted for publication at this stage.
The authors declare that this paper has not been published and is not under consideration for publication elsewhere. We also have no conflicts of interest to disclose.
Thank you for your consideration!
Yours sincerely,
Ana Carolina Abrantes
Corresponding author
REVIEWER 1:
Dear authors,
A good work in general, but some aspects should be reviewed:
- Lines 29 to 32: Could be interesting to consider the citation of https://doi.org/10.4195/nse.2013.0024
Answer: The authors are grateful for the suggestion and have included the article in their work.
- Lines 78 to 84:
- Is there 206 surveyed hunters enough to consider this study representative to the hunter population in Portugal? How many people practice hunting in Portugal?
- Please provide a complete copy of the questionnaire, no only a summary, as shown in Table 1 (maybe as an Annex). To make possible the replicability of your survey.
- Have you consult other previous surveys to elaborate your questionnaire?
- How did you avoid poll crashing in the online version?
Answer: The authors are grateful for the questions. It is estimated that there are around 300,000 hunters in Portugal, but there is no indication of how many are dedicated to wild boar hunting. As the population of hunters is also ageing, it is difficult to obtain voluntary questionnaires. The focus of the work was on hunters who consume their own wild boar meat and this percentage is not only small but also unknown.
The questions in the table are exactly the translation of the questionary written in Portuguese based on other surveys already realised in other European countries and distributed to Portuguese hunters. The authors are willing to translate the entire questionnaire into English in the original prototype if the reviewers and editor deem it indispensable.
The authors avoided pool crashing since all the questionnaires delivered on paper had previously been asked if they had already completed the online questionnaire, and the online version had a similar question to the paper version.
- Lines 100-101: "Two hundred and six hunters (n=206) have responded to the survey, from all over Portugal" Could you provide a map with the origin of the answers to assure this.
Answer: Yes, we ask in the survey the origin region of the hunters. This information was added to the manuscript.
- Lines 103-104:"In all of the 206 responses, the initial examination by a veterinarian is also not a common practice". Indicate a percentage.
Answer: The focus of the work was on hunters who self-consume their own wild boar and this practice is mostly carried out in areas of Portugal where the initial examination by a veterinarian is not obligatory, so the percentage of these hunters is relative. It's 100% when we say that all hunters don't observe an initial examination by a veterinarian on the hunts they go on. However, the percentage drops to around 95% when it comes to whether these hunters have ever self-consumed wild boar meat that had previously been analysed by a veterinarian.
The authors have clarified the issue and rewritten the manuscript.
- Line 120: Could be convenient to add a reference closer to the geographic area of the survey https://doi.org/10.1111/zph.12343
Answer: The reference has been added to the suggested sentence/line. This reference was already listed as 22.
- Lines 171-175: The trained person course it is mentioned. Are there differences in the answers provided by hunters who have completed the course in comparison with those who haven't completed it?
Answer: The number of replies from hunters who have already taken the initial examination course to become a "trained person" who responded to the survey is minimal. So, you can't make a statistical analysis to draw that conclusion. However, descriptive analysis does suggest that there are better practices by those who have taken the course.
It should be noted that in Portugal, of the 300,000 hunters, only 300 will have completed this course by September 2023.
- Line 182: [17,34-35]"." A dot "." is lacking.
Answer: The error has been corrected.
- Line 188: "obtained in various scientific studies published with data from Portugal [17,34-35]". Reference 35 is not a scientific study, but a guide. And it comes from Spain, with no mention about Portugal in the text.
Answer: The authors would like to thank you for your comments. There was a problem with references 35 and 36, as reference 35 was missing from the previous version submitted to the editor. This was a study on toxoplasma in Portugal and reference 36 on by-products. The error in the manuscript has been corrected.
-Lines 189-197: It is interesting to add a comparison with the results achieved by https://doi.org/10.3390/foods11030368
Answer: The authors are grateful for the suggestion and have included the article in their work.
- Line 197: There is a reference, identified as 36, that does not appear in the bibliography.
Answer: The authors would like to thank you for your comments. There was a problem with references 35 and 36, as reference 35 was missing from the previous version submitted to the editor. This was a study on toxoplasma in Portugal and reference 36 on by-products. The error in the manuscript has been corrected.
- Line 287: Problem with the format in reference 17.
Answer: The error has been corrected.
Conclusions (lies 226 to 237): some conclusions are similar to those exposed by https://doi.org/10.3390/foods11030368 so should be cited.
Answer: The authors are grateful for the suggestion and have included the article in their work.
Reviewer 2 Report
Comments and Suggestions for Authors
Title: i suggest that the title be changed to read as follows: "Knowledge and Hygiene practices among Portuguese hunters who practice self-consumption of wild boars"
Abstract: there is a need to improve on the abstract. Details are provided in the soft copy attached.
Introduction: You concentrated mainly on bacterial contamination of carcasses of wild animals. You need to touch on what we know about the knowledge of hunters and their hygiene practices. You also need to clearly show the gap that your study seeks to address.
Materials and Methods: You said that you randomly selected participants, but at the same time you say that you wanted to reach as many as possible? This does not make sense. Unless you wanted to reach as many as possible but only managed to get only 206. If that is the case then you should make it clear. Otherwise as the section stands it is not clear how you sampled and as a result it would be impossible to reproduce your work, which is a key indicator for a good manuscript.
Ethical aspects: were the participants given the oportion of withdrawing from the study and were the objectives clearly explained to them? That needs to be made clear.
Results: this is the lowest point of this study. You basically provided roughly 10 lines of summary of the results. You need to describe your results in text. A table and a one liner saying "The total percentage show in Table 1 are of the effective responses, having excluded the null responses that may occur in the surveys distributed" is not sufficient". A detailed description of all the results in the table is needed. May be you should consider naming the section results and discussion so that you have only one section instead of Results and Discussion.
Furthermore, only one table for a research article is too small. This qualifies perharps to be published as a letter.
Discussion: I suggest that have one section that combines the results and the disccussion.
line 114 to 117 you talk of problems that relate to hygiene that might occur. however, you do not reference your results. or is this from literature?
line 117-119 The lack of initial examination of carcasses poses a health risk not only to hunters but to a wide range of 118 people (family, friends, etc.), who probably have scarce information about risks involved 119 in large game meat consumption [19]. Before you say this i suggest that you first cite your results. what did you see and then tell us what others have reported and the significance of your findings.
You need to the above for all the results -the results in the table should appear in the text.
Conclusion: See comments in the attached article.
References: You also needed to cite articles on what we know about the knowledge and hygiene practices of hunters. why is hunters having good knowledge and adotpting good hgyiene practices important?

The quality of English is very poor. The authors can hardly or bearly express themselves in English. please refer to the article for details of this. Definitely a professional English language editor should be engaged if the paper is to be published.
Author Response
Ana Carolina Abrantes
University of Trás-os-Montes and Alto Douro, (UTAD)
Quinta de Prados
5001-801 Vila Real, PORTUGAL
carolina.psca@gmail.com
Editor of Zoonotic diseases Journal
Vila Real, September 14th 2023
Please find enclosed the resubmission of the manuscript entitled “Knowledge and Hygiene Practices among Portuguese hunters who practice self-consumption of wild boars” renamed as a reviewer’s suggestion.
After an exhaustive reformulation of the text and improvement of the English, we think that the document is now more clarified.
I would add that all the questions raised by the reviewers were answered and the text was adjusted in order to clarify the confusing ideas that were exposed. Attached is a document with individual responses to each point pointed out by the reviewers.
Thank you in advance for the effort of everyone and especially the reviewers for the comments, we hope that the joint effort has substantially improved the document in order to gather conditions to be accepted for publication at this stage.
The authors declare that this paper has not been published and is not under consideration for publication elsewhere. We also have no conflicts of interest to disclose.
Thank you for your consideration!
Yours sincerely,
Ana Carolina Abrantes
Corresponding author
REVIEWER 2:
title: i suggest that the title be changed to read as follows: "Knowledge and Hygiene practices among Portuguese hunters who practice self-consumption of wild boars"
Answer: The authors are grateful for the suggestion and have changed the title as suggested.
Abstract: there is a need to improve on the abstract. Details are provided in the soft copy attached.
Answer: The authors are grateful for the suggestion and have changed the abstract as suggested.
As questioned, driven hunt means an organized act of hunting where there are dogs looking for game animals (wild boar, wild boar...) and the hunters stand at specific posts waiting to observe these animals and be able to hunt them.
Introduction: You concentrated mainly on bacterial contamination of carcasses of wild animals. You need to touch on what we know about the knowledge of hunters and their hygiene practices. You also need to clearly show the gap that your study seeks to address.
Answer: The authors would like to thank you for your comment and have rewritten the paper to update this information.
Materials and Methods: You said that you randomly selected participants, but at the same time you say that you wanted to reach as many as possible? This does not make sense. Unless you wanted to reach as many as possible but only managed to get only 206. If that is the case then you should make it clear. Otherwise as the section stands it is not clear how you sampled and as a result it would be impossible to reproduce your work, which is a key indicator for a good manuscript.
Answer: The study period was short and since most hunters in Portugal are of a certain age and it's difficult to reach them online, many of the surveys were done on paper, so although we tried to include as many responses as possible, 206 were possible in the defined time period. This information has been included in the manuscript.
Ethical aspects: were the participants given the oportion of withdrawing from the study and were the objectives clearly explained to them? That needs to be made clear.
Answer: When the survey was distributed (online and paper versions), the participants were informed of the aim of the study and the final product of the work (which is this manuscript). All participants completed an informed consent form. No personal data was obtained from the participants.
Results: this is the lowest point of this study. You basically provided roughly 10 lines of summary of the results. You need to describe your results in text. A table and a one liner saying "The total percentage show in Table 1 are of the effective responses, having excluded the null responses that may occur in the surveys distributed" is not sufficient". A detailed description of all the results in the table is needed. May be you should consider naming the section results and discussion so that you have only one section instead of Results and Discussion.
Furthermore, only one table for a research article is too small. This qualifies perharps to be published as a letter.
Discussion: I suggest that have one section that combines the results and the disccussion.
Answer: The authors are grateful for the suggestion. The choice to only use the table to mirror the results came from the editorial proposal to avoid repeating the subject constantly.
The authors will follow the suggestion to combine results + discussion.
line 114 to 117 you talk of problems that relate to hygiene that might occur. however, you do not reference your results. or is this from literature?
Answer: The authors have added the references in the marked sentence in order to improve the engagement between the results of this article and the literature.
line 117-119 The lack of initial examination of carcasses poses a health risk not only to hunters but to a wide range of 118 people (family, friends, etc.), who probably have scarce information about risks involved 119 in large game meat consumption [19]. Before you say this i suggest that you first cite your results. what did you see and then tell us what others have reported and the significance of your findings.
Answer: The authors added their results.
You need to the above for all the results -the results in the table should appear in the text.
Answer: The authors are grateful for the suggestion. The choice to only use the table to mirror the results came from the editorial proposal to avoid repeating the subject constantly. The authors will follow the suggestion to combine results + discussion and express the results in this text.
Conclusion: See comments in the attached article.
Answer: The authors are grateful for the suggestion and have changed the abstract as suggested.
References: You also needed to cite articles on what we know about the knowledge and hygiene practices of hunters. why is hunters having good knowledge and adotpting good hgyiene practices important?
Answer: The list of references has been updated to cover the shortcomings that the reviewers noted in the work.

Reviewer 3 Report
Comments and Suggestions for Authors
This manuscript is very interesting because very little is known about hunters’ evisceration and handling practices affecting the game meat safety and quality. The authors showed that there are huge differences in hygiene practices and in meat inspection knowledge.
What are the training requirements for trained hunters regarding hygiene and safety in Portugal? There is a specific course (line 171) but is there a compulsory test, which must be passed for trained hunters? How many of the hunters were trained and should have sufficient competence to perform the initial inspection?
Pity, that the authors have not used statistical analysis to interpret their results. The variables are categorical and it would have been easy to test possible associations between the variables with multivariate analyses. It is impossible from the table to see if there is any association/correlation between different question: e.g., has the same hunter answered that fecal contamination is removed by washing (question 3) and fecal contamination is removed without water (question 4).
No questions about trichinella testing and brucella was included in the study, why? Is Erysipelothrix a common finding in wild boars?
In the title, is the word “literacy” correct? Do you mean “knowledge”? I am also not sure if the title is written in correct English: “in wild boar self-consumption zoonotic risk practices”?
In the results part, the authors only present the results in a summary table. I recommend to shortly present the (most important) part of the table also in the text. Authors have presented lot of results in the discussion part, which I suggest being moved to the results part. Another alternative is to combine the results and discussion.
In the table point 3: “Lage (?) game initial examination course”? Is this course compulsory for trained hunters?
Author Response
Ana Carolina Abrantes
University of Trás-os-Montes and Alto Douro, (UTAD)
Quinta de Prados
5001-801 Vila Real, PORTUGAL
carolina.psca@gmail.com
Editor of Zoonotic diseases Journal
Vila Real, September 14th 2023
Please find enclosed the resubmission of the manuscript entitled “Knowledge and Hygiene Practices among Portuguese hunters who practice self-consumption of wild boars” renamed as a reviewer’s suggestion.
After an exhaustive reformulation of the text and improvement of the English, we think that the document is now more clarified.
I would add that all the questions raised by the reviewers were answered and the text was adjusted in order to clarify the confusing ideas that were exposed. Attached is a document with individual responses to each point pointed out by the reviewers.
Thank you in advance for the effort of everyone and especially the reviewers for the comments, we hope that the joint effort has substantially improved the document in order to gather conditions to be accepted for publication at this stage.
The authors declare that this paper has not been published and is not under consideration for publication elsewhere. We also have no conflicts of interest to disclose.
Thank you for your consideration!
Yours sincerely,
Ana Carolina Abrantes
Corresponding author
REVIEWER 3:
This manuscript is very interesting because very little is known about hunters’ evisceration and handling practices affecting the game meat safety and quality. The authors showed that there are huge differences in hygiene practices and in meat inspection knowledge.
What are the training requirements for trained hunters regarding hygiene and safety in Portugal? There is a specific course (line 171) but is there a compulsory test, which must be passed for trained hunters? How many of the hunters were trained and should have sufficient competence to perform the initial inspection?
Answer: There is a specific course in Portugal for hunters to be considered a "trained person" to carry out the initial examination of the shot game. This course has been approved by the Portuguese authorities, with a theoretical and practical component and an assessment in accordance with European regulations. A trained person cannot carry out an official inspection of the carcasses, but rather an initial examination that allows the carcasses to be sent gutted and headless to slaughterhouses and game preparation establishments. They are responsible for the first observation of the sanitary condition of the carcasses on site.
Pity, that the authors have not used statistical analysis to interpret their results. The variables are categorical and it would have been easy to test possible associations between the variables with multivariate analyses. It is impossible from the table to see if there is any association/correlation between different question: e.g., has the same hunter answered that fecal contamination is removed by washing (question 3) and fecal contamination is removed without water (question 4).
Answer: In a statistical context, the associations found were weak or not statistically significant. All questions are independent, and in the case of co-related questions, "if not" or "if yes" was placed at the beginning of each question so that there would be no overlapping of answers for the same subject.
No questions about trichinella testing and brucella was included in the study, why? Is Erysipelothrix a common finding in wild boars?
Answer: The questionnaire only included questions about lesions that can be found during the initial examination of wild boar carcasses and that are pathognomonic for a particular disease. Both Brucellosis and Trichinella can only be catalogued as diseased animals if laboratory tests are carried out, although brucellosis has non-specific signs.
However, a study on hunters' knowledge of trichinella has already been published in Portugal: https://doi.org/10.1111/zph.12800
In the title, is the word “literacy” correct? Do you mean “knowledge”? I am also not sure if the title is written in correct English: “in wild boar self-consumption zoonotic risk practices”?
Answer: The authors understood the confusion and, with the suggestion of another reviewer, changed the title to “"Knowledge and Hygiene practices among Portuguese hunters who practice self-consumption of wild boars”.
In the results part, the authors only present the results in a summary table. I recommend to shortly present the (most important) part of the table also in the text. Authors have presented lot of results in the discussion part, which I suggest being moved to the results part. Another alternative is to combine the results and discussion.
Answer: The authors are grateful for the suggestion. The choice to only use the table to mirror the results came from the editorial proposal to avoid repeating the subject constantly. The authors will follow the suggestion to combine results + discussion as two reviewers suggest.
In the table point 3: “Lage (?) game initial examination course”? Is this course compulsory for trained hunters?
Answer: For a hunter to be considered a trained person to carry out the initial examination correctly and with authorisation from the authorities, it is mandatory to have attended this course designated by European regulation.

Round 2
Reviewer 1 Report
Comments and Suggestions for Authors
After the changes introduced by authors, I consider the research ready to be published.
Author Response
After an exhaustive reformulation of the text and improved English, we think the document is now more clarified.
Thank you in advance for the effort of the reviewers.
Reviewer 2 Report
Comments and Suggestions for Authors
The authors have not used a qualified English language practitionar to assist with improving on how the manuscript reads. The English is very poor and as a result I am unable to review. I find myself having to edit the language for the authors which i think is not the job of the reviewer.
I have suggested some editorial changes, but that is not all. I was not able to go through the paper this time around because the language is just too poor.
Comments on the Quality of English LanguageIt is obvious that English is not the first language of the authors. So there language is too poor to be able to make sense of what authors are saying in most instances.
Author Response
Vila Real, September 25th 2023
Please find enclosed the resubmission of the manuscript renamed “Hygiene knowledge and practices of Portuguese hunters using wild boar meat for private consumption".
After an reformulation of the text and improvement of the English, we think that the document is now more clarified.
Thank you in advance for the effort of the reviewer for the comments.